# Experimental Investigation on Near-Field Acoustic Propagation Characteristics of Leakage Detection in Submarine Pipelines

**Kang Zhang \*, Ruize Ma** **, Tao Geng, Jiannan Yang and Yongjun Gong**

College of Naval Architecture and Ocean Engineering, Dalian Maritime University, Dalian 116026, China; maruize@dlmu.edu.cn (R.M.); gt970615@dlmu.edu.cn (T.G.); yjn0525@dlmu.edu.cn (J.Y.); gyj@dlmu.edu.cn (Y.G.)
\* Correspondence: zhangkang@dlmu.edu.cn

**Abstract:** The leakage of subsea oil and gas pipelines can have adverse impacts on production progress and the ecological environment. Investigating the sound source and near-field sound propagation of pipeline leaks is essential for understanding the acoustic characteristics of and variations in these leaks. Such understanding is significant for the accurate detection and location of small leaks in pipelines. In this study, we designed an experimental system to study the characteristics of leakage sound signals. We introduced the formation mechanism of leakage sound sources and reviewed corresponding theoretical research. The leakage sound signal's characteristic frequency range was determined to be between 1 kHz and 2 kHz. We examined the effects of pipeline pressure, leakage aperture, and detection distance on the acoustic signal characteristics. The results show that as internal pipe pressure increases, the leakage sound signal intensity first increases and then decreases. As the leakage aperture increases, the intensity of the leakage sound signal increases. Within a short distance, the intensity remains consistent regardless of detection distance. The results of this experimental study can guide the acoustic internal detection of pipelines. This study has practical significance in the timely detection of small leaks in pipelines and preventing leakage accidents from occurring.

**Keywords:** pipeline leakage internal detection; near sound field; sound propagation characteristics; experimental method

## 1. Introduction

Offshore oil and natural gas occupy a very important position in modern industry [1]. Pipelines, central to oil and gas transport, are essential in the development and production of these resources [2]. Over time, as the pipeline operates, leakages can occur due to corrosion, aging, cracks, natural leaks, and other reasons [3]. Such leakages not only affect normal production activities but also cause serious economic losses and environmental degradation [4]. Therefore, an effective leak detection method is of great significance for pipeline health detection.

Leak detection techniques can generally be classified based on their application, either inside or outside the pipeline [5,6]. Over the years, numerous systems for detecting pipeline leakage have been developed, such as the magnetic flux leakage internal detection method, the negative pressure wave method, the ultrasonic internal detection method, and the distributed fiber optic method [7–10]. Of these, acoustic detection technology is particularly notable due to its high sensitivity, precise positioning, lower false alarms, quick response, and adaptability. It has been widely used in pipeline systems for decades [11]. The acoustic listening detection method is among the earliest acoustic leak detection methods [12]. This method employs devices equipped with sensors or leak detectors to collect sound signals. These signals are then amplified and relayed to staff through headphones to determine the presence of leaks. However, this method relies on the operator's auditory sensitivity and experience, making it somewhat subjective and restrictive. As sensor technology and

signal processing advanced, Fuchs et al. designed a correlation instrument consisting of a central host and two sensor slaves [13]. The instrument operates by placing two induction probes at different endpoints of a singular pipeline. It then measures the time taken for a signal to reach sensors on both sides of the pipeline. From the distance between the two slave machines and the speed at which a leakage signal propagates within the pipeline, the precise leak location can be deduced. A disadvantage of this method is the relatively high cost of the associated instruments, combined with challenges in underground deployment and maintenance. In case of damage, remedial actions are cumbersome. Over time, traditional acoustic detection methods have seen improvements and refinements, leading to the development of pipeline acoustic internal detection [14]. This modern approach involves installing sensors on the pipeline detector, which moves or swims within the pipeline, making this method capable of detecting very small leaks. A notable example of an in-pipe sensor system is the PIG (Pipeline Inspection Gauge)-based leak detection method [15]. This system utilizes the PIG's movement along the pipeline's flow, with an attached hydrophone gathering leak sound signals from within the pipeline. While PIGs move through the pipeline, they generate more noise due to the friction between their wheels and the pipeline wall. This leads to a low signal-to-noise (SNR) ratio. Due to their tight fit against the pipe wall, similar to a piston, PIGs are vulnerable to problems caused by pipe deformation and bending. These issues can result in blockage and restrict their use in subsea pipelines [16]. In response to these challenges, a new free-swimming, unconstrained acoustic leak detector known as SmartBall has been invented [17]. Compared with traditional cylindrical PIGs, SmartBall has a spherical shape and rolls through the pipeline during its operation. This design significantly reduces the noise generated by equipment movement within the pipeline. Moreover, with a diameter smaller than the pipeline's inner diameter, SmartBall offers superior possibilities, making it more suitable for long-distance subsea pipelines due to its reduced risk of blockages.

The hydrophone's proximity to the leak point allows for precise leak detection, minimizing interference from detector motion noise. This design enhances sensitivity, especially to smaller leaks. The noise produced by a leaking pressure pipeline can propagate through both the pipe wall and the fluid inside the pipeline. The frequency components of the leakage signal encompass both low- and high-frequency elements [18]. Papastefanou et al. devised an experimental configuration to investigate the underlying physical mechanism behind leakage noise. Their findings suggest that turbulence occurring in water jets is likely the predominant source of leakage noise in plastic water pipes [19]. Xiao et al. investigated the characteristics of leakage noise resulting from variations in leak shapes, sizes, and pressures. They concluded that leakage noise is mainly concentrated in low frequencies and can be transmitted between branches and bends. The research suggests that leak noise is mainly affected by the leak area, while the leak's shape has minimal impact [20]. Hunaidi and Chu conducted a study on the acoustic frequency characteristics of leakage signals specifically in PVC distribution pipes [21]. Their analysis encompassed the frequency content of sound or vibration signals, assessing variables such as leakage type, flow rate, pipeline pressure, and seasonal variations. Their study also explored the shifts in attenuation rate and propagation speed across different frequencies. Meanwhile, Xu et al. studied the principles of sound generation and the characteristics of sound sources related to pipeline leakage, simulating the leakage sound field based on aerodynamic acoustics. Secondly, a laboratory test method for recognizing acoustic signals from natural gas pipeline leaks was proposed, using wavelet packet transform and fuzzy support vector machine pattern classification [22]. While there have been numerous studies on the characteristics of pipeline leakage sound sources and the propagation of such sounds, there is limited research specifically on the characteristics of near-field leakage sounds in pipelines, especially within the context of SmartBall applications.

This paper aims to investigate the characteristics of pipeline leakage noise sources and the propagation properties of such noise in the near field. We evaluate factors such as leak pressure, leak aperture, and the distance between the leak aperture and the hydrophone to

understand their effect on the leak sound signal. The structure of this paper is organized as follows: a theoretical background of the leakage noise source, an introduction to the experimental system, methods of data processing, analysis and discussion of the results, and the conclusion.

## 2. Research on Leakage Sound Sources

### 2.1. Composition of Leakage Noise Sources

Research by [23] has identified three primary sound sources resulting from fluid leakage in pressure vessels: the first is turbulent sound, where the pressure difference between the inside and outside of the pipeline can cause high-speed jets containing a large number of vortices, forming turbulent sound. This jet, in itself, becomes a sound source void sound: near the leakage point, a localized low-pressure zone emerges. When this local pressure drops below the air separation pressure, the dissolved gas in the liquid escapes, forming bubbles in the leakage point's low-pressure area. The bursting of these bubbles produces a cavitation sound. There is also fluctuating pressure in the turbulent boundary layer. The turbulent boundary layer is characterized by its time and space randomness. Random-velocity disturbances within this layer generate random pulsating pressure sounds. Of these three sound sources, turbulent sound is the predominant source of leakage noise.

### 2.2. Turbulent Sound Formation Process

When a pipeline leaks, the fluid sprays out through the leakage hole, forming a boundary layer adjacent to the wall of the leakage hole. This boundary layer inception serves as the starting point of turbulent sound. Within this layer, factors such as velocity shear and anisotropy cause an uneven fluid velocity distribution, accompanied by velocity fluctuations. This instability promotes the formation of small-scale vortices. These vortices, subjected to external and internal forces such as fluid shear and turbulent energy dissipation, lead to the growth and interaction of these vortices, forming larger vortex structures. Once these vortices achieve a certain scale, they might detach from the wall. These detached, large vortices undergo stretching and twisting, forming severe velocity alterations. As they move further from the wall, they diminish, fragmenting into smaller vortices. These smaller vortices, evolved from the larger ones, continually form and disappear. During this vortex detachment and evolution, there are local shifts in pressure and velocity. These changes form pressure waves within the fluid, resulting in turbulent sound. Continuous leakage fuels this vortex evolution process, thereby producing continuous leakage noise. Figure 1 illustrates the turbulent sound formation process.

### 2.3. Lighthill Acoustic Analogy Method

When a pipeline experiences a leak, the pressure disparity between the internal and external sections of the pipeline can trigger turbulent jets (see Figure 1). The primary origin of leakage noise is attributed to the turbulence surrounding the leak [24]. In 1952, Lighthill proposed the theory of acoustic analogy. This theory transforms acoustic phenomena into fluid dynamics problems, facilitating a deeper understanding of sound propagation mechanisms and sound source characteristics. The Lighthill acoustic analogy is based on the N-S (Navier–Stokes equations) mass conservation equation and the N-S momentum conservation equation:

Mass conservation equation:

$$\frac{\partial \rho}{\partial t} + \frac{\partial \rho v_i}{\partial x_i} = 0 \tag{1}$$

Momentum conservation equation:

$$\frac{\partial \rho v_i}{\partial t} + \frac{\partial \rho v_i v_j}{\partial x_j} = -\frac{\partial p_{ij}}{\partial x_j} \tag{2}$$

After transformation, the following Lighthill equation can be written as follows:

$$\frac{\partial^2 \rho}{\partial t^2} - c^2 \nabla^2 \rho = \frac{\partial^2 T_{ij}}{\partial x_i \partial x_j} \tag{3}$$

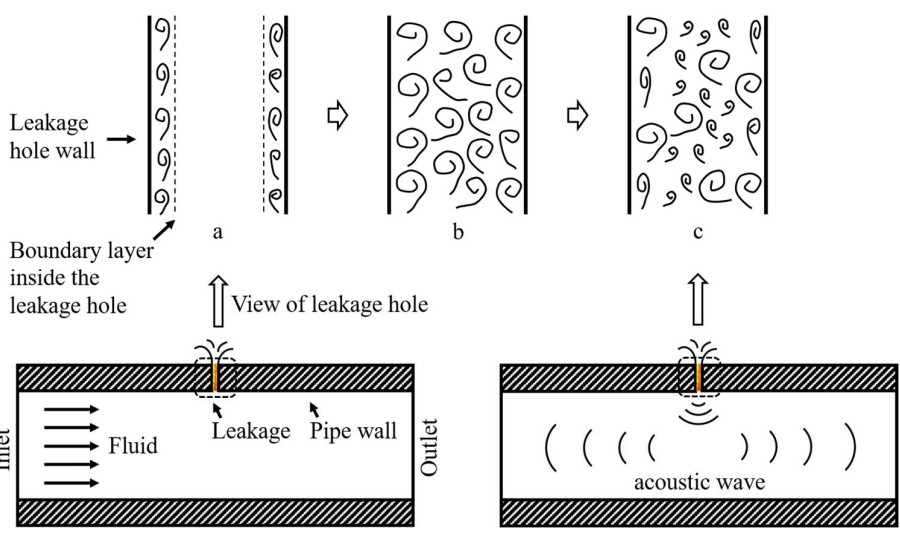

**Figure 1.** Turbulent sound formation process. (**a**–**c**): present the different flow state at different flow velocity (from low to high).

The Lighthill equation [25] separates the terms related to sound sources and sound propagation: with the left side of the equation, there is a decoupling of an acoustic operator, whereas the right side represents the equivalent source terms.

Here, $T_{ij} = \rho v_i v_j + p_{ij} - c^2 \rho \delta_{ij}$ (i, j = 1, 2, 3). $T_{ij}$ denotes the Lighthill turbulent stress tensor; $\rho$ represents the fluid density; c denotes the acoustic velocity; $x_i$, $x_j$ signify the Cartesian coordinates; $\delta_{ij}$ represents the Kronecker symbol (when the subscripts are equal, the value is 1, and when the subscripts are not equal, the value is 0); $p_{ij}$ denotes the sound pressure; $v_i$, $v_j$ denote the fluid velocity in the corresponding x-direction; and t represents the time.

Ffowcs, Williams and Hawkings [26] applied generalized functions to solve the problem of sound generation by moving objects in fluids and obtained the FW-H (Ffowcs Williams–Hawkings) equation:

$$\frac{\partial^2 \rho}{\partial t^2} - c^2 \nabla^2 \rho = \frac{\partial}{\partial t}\left[\rho u_i \frac{\partial f}{\partial x_i}\delta(f)\right] - \frac{\partial}{\partial x_i}\left[(p\delta_{ij})\frac{\partial f}{\partial x_i}\delta(f)\right] + \frac{\partial^2 T_{ij}}{\partial x_i \partial x_j} \tag{4}$$

where $\delta(f)$ represents the Dirac function, $u_i$ is the component of the fluid velocity in the direction $x_i$ (i = 1, 2, 3), and *p* is the compressive stress tensor. The three items on the right are sound source terms: monopole sound source, dipole sound source, and quadrupole sound source. When a pipeline leaks, the fluid spews out at high speed from the leakage hole and then forms a high-speed jet area inside the leakage hole. The monopole sound source caused by mass flow can be ignored. The FW-H equation is in good agreement with the leakage far sound field. For the leakage in the pipeline, the near-sound-field Lighthill equation is more suitable.

The Möhring equation [27] can accurately calculate nonlinear sound fields and sound propagation in non-uniform media and can be used to extract equivalent sound sources in high Mach number and high Reynolds number flow fields. The surface sound source term ($R_s$) and volume sound source term ($R_v$) in the Möhring equation correspond to the dipole sound source and quadrupole sound source in the leakage sound field, respectively. The Möhring equation is as follows:

$$R_v = \int \frac{\partial N_a}{\partial x} FFT \left[ \frac{\rho}{\rho_T} \left( vw - \frac{\partial \tau_{ij}}{\partial x_j} - \frac{u}{\rho_T} \frac{\partial \rho}{\partial s} \right)_p \frac{\partial s}{\partial t} + \frac{\rho T}{\rho_T} \frac{\partial s}{\partial x} \right] dV + i\omega FFT \left\{ \int \left[ \frac{N_a}{\rho_T} \left( \frac{\partial \rho}{\partial s} \right) \frac{\partial s}{\partial t} - \frac{\rho u N_a}{\rho_T^2} \frac{\partial \rho_T}{\partial x} \right] dV \right\} \quad (5)$$

$$R_S = i\omega FFT \left( \int \frac{\rho u \cdot n_i}{\rho_T} dS \right) \quad (6)$$

Here, $N_a$ represents the shape functions of finite elements; $\rho$ and $\rho_T$ denote the static density and stagnation density, respectively; u, v, w represent the velocity components in the x, y, z directions, respectively; $\tau_{ij}$ denotes the viscous stress tensor; s stands for entropy; $\omega$ stands for angular velocity; $n_i$ is the integral surface's normal unit vector; and T denotes the temperature. The terms related to $\frac{\partial \tau_{ij}}{\partial x_j}$, $\frac{\partial s}{\partial t}$, $\frac{\partial \rho_T}{\partial x_i}$ represent viscous effects, entropy changes, and density fluctuations, respectively; S signifies the area, while V indicates volume. FFT is the fast Fourier transform compared to the Lighthill equation, and the Möhring equation is more suitable for high Mach number (M > 0.2) situations. However, concerning the acoustic problem of pipeline leakage studied in this article, the Mach number is relatively low, as inferred from experimental data. Therefore, the Lighthill equation is deemed more suitable for this study.

## 3. Experimental Setup

### 3.1. Leakage Pipeline System

The experimental setup (see Figure 2) consists of components such as a pipe, interchangeable leak holes, a flowmeter, a pressure gauge, a pulsation damper, a back-pressure valve, and a water pump. This pipeline segment features six swappable leak holes alongside ports for hydrophone installation. To reduce the effects of sound wave reflections on the experiment, sound-absorbing materials line the inner walls of the pipes at both ends. Additionally, vibration-dampening materials are situated between the pipe and its mounting rack to minimize external interference to the system. Designed for easy substitution, the replaceable leak holes come with external threading, facilitating swift changes at specific leak points within the pipeline. The water feed system for the pipe is bifurcated into two routes, both regulated by ball valves. The primary route integrates a flowmeter, pressure gauge, pulsation damper, and ball valve in sequential arrangement. This setup permits the accurate measurement of key parameters during testing, with the pulsation damper ensuring reduced pressure inconsistencies within the pipe. The secondary route, comprising water pipes and ball valves, is optimized for swift water supply during testing. Connected to the pump's exit is the back-pressure valve, which then links to the pipeline's water supply system. This valve is crucial for modifying the internal pressure of the system as needed during the experiment. To diminish the influence of pump-induced vibrations on the study, rubber hoses are used to bridge the pump and the pipeline's water supply system.

### 3.2. Measurement Technique

To minimize the influence of external variables on the experiment, the leakage test was conducted in an indoor environment. A hydrophone was installed within the experimental pipeline to capture the acoustic signals generated by leakage. Before initiating the experiment, a test leak hole was positioned at the designated location on the pipeline. To safeguard the instrumentation from potential damage due to strong transient pulsations upon starting the water pump, the upper instrument inlet passage was closed using a ball valve, while the lower inlet passage was opened. During the experiment, the water pump was activated, and the desired pressure level was adjusted using a back-pressure valve. After stabilizing the internal pipeline pressure, the upper inlet passage was opened and the lower one was closed. Pressure and fluid state parameters inside the pipeline were measured using a pressure gauge and flowmeter, respectively. The acoustic signals captured by the hydrophone were collected and analyzed using computer software. Upon completing

the measurements, the lower inlet passage was opened, the upper inlet passage closed, the water pump deactivated, and the back-pressure valve adjusted to release internal pipeline pressure. Lastly, the test leak hole was sealed with a plug. A schematic diagram of the experimental setup is provided in Figure 2.

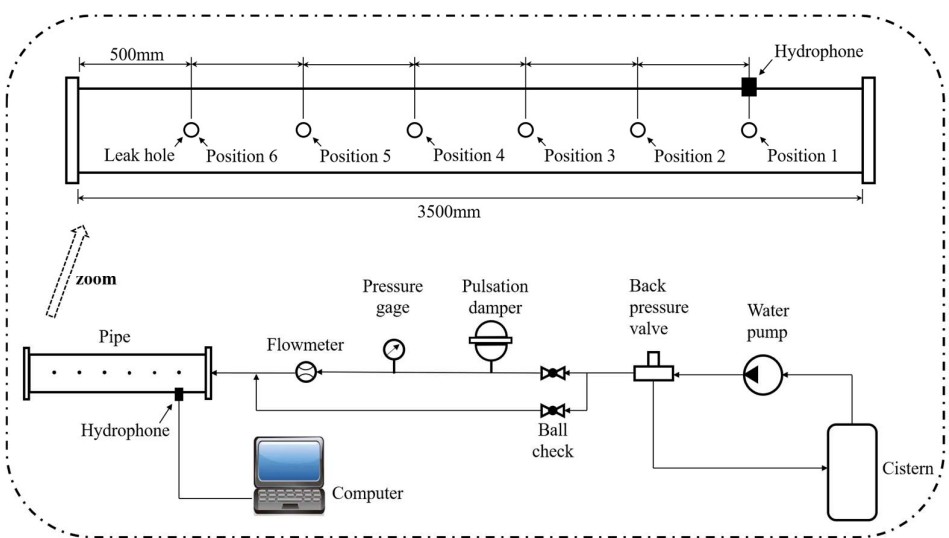

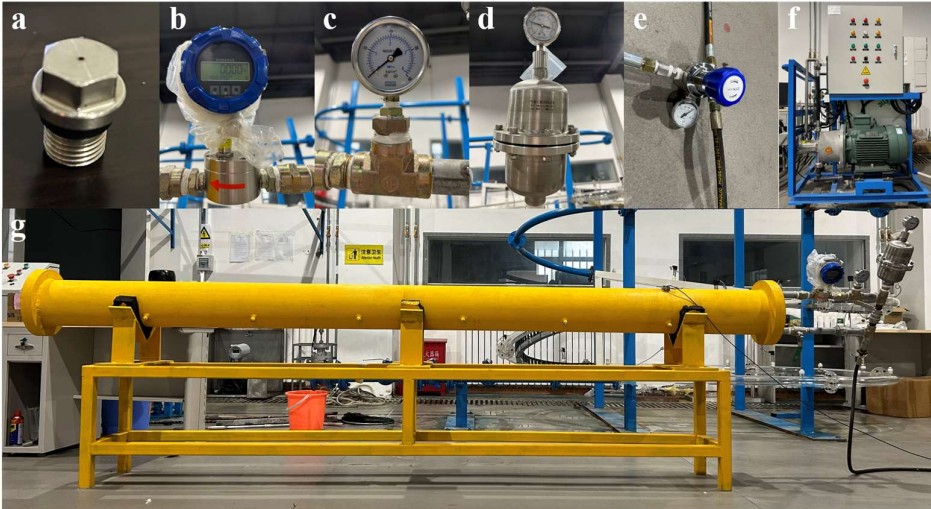

**Figure 2.** Schematic diagram and physical diagram of leakage experiment system: (**a**) leak hole; (**b**) flowmeter; (**c**) pressure gauge; (**d**) pulsation damper; (**e**) back-pressure valve; (**f**) water pump; (**g**) experimental pipeline.

### 3.3. Experimental Condition

An iron pipe with a straight design, 200 mm in diameter and 3500 mm in length, was selected as the test pipeline. Both ends of the pipe are fitted with flanges and feature six leakage holes and ports for hydrophone installation (refer to Figure 2). These leakage holes are designed as bolts, allowing for versatile installation at predetermined positions along the pipeline. The hole diameters are 0.5 mm, 1.0 mm, 1.5 mm, 2.0 mm, and 2.5 mm. The experimental setup requires a constant pressure supply, which is maintained using a high-pressure water pump. Essential equipment for this experiment comprises a hydrophone, flowmeter, pressure gauge, pulsation damper, and back-pressure valve. The hydrophone, pivotal to this experiment, detects leakage sound signals when installed within the pipeline (see Figure 3). Detailed specifications for the hydrophone can be found in Table 1. All the devices are interconnected, forming a comprehensive leakage pipeline experimental system.

Due to the operational limits of the instruments, the system's peak working pressure is set at 1 MPa, with a maximum flow rate of 240 L/h.

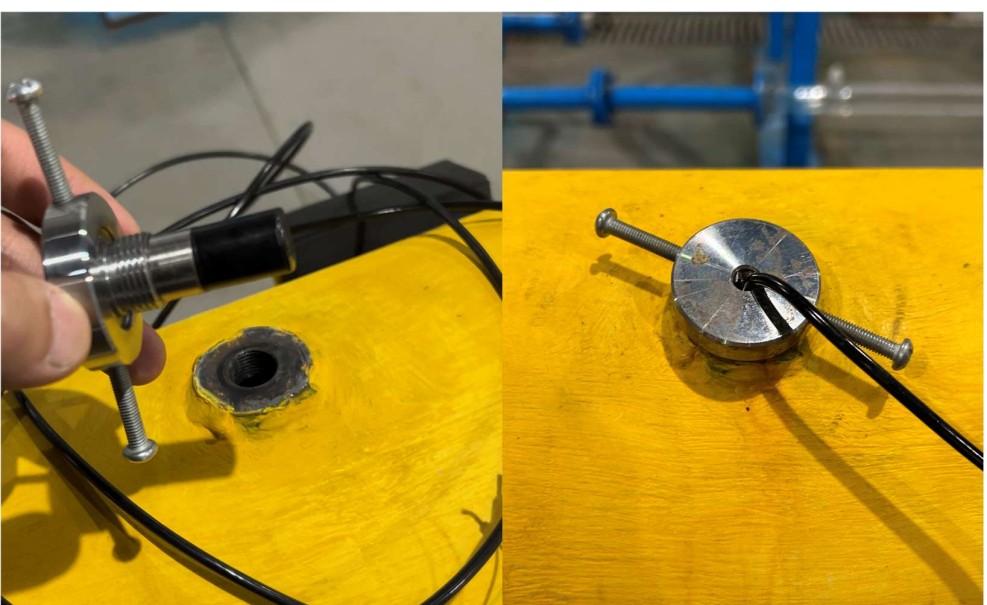

**Figure 3.** Hydrophone and its installation methods.

**Table 1.** Hydrophone performance index; measuring temperature = 25 °C.

| Parameter | Symbol | Performance | Unit |
|:---:|:---:|:---:|:---:|
| Dynamic range | FS | $\pm 1000$ | Pa |
| Sound sensitivity | $S_P$ | $-185 \pm 3.0$ | dB re 1 V/$\mu$Pa |
| Nonlinear | | $<1$ | % |
| Noise resolution (1 kHz) | $N_n$ | $\sim 60$ | dB re $\mu$Pa/$\sqrt{}$Hz |
| Bandwidth | BW | 20 to 20,000 | Hz |
| Electrical characteristic | | | |
| Built-in amplifier gain | $A_V$ | 40 | dB |
| Maximum output amplitude | $V_o$ | 1 | $V_{PPK}$ |
| Current | $I_{rms}$ | 4 | mA |
| Directivity | | all | |
| Operating depth | $D_{op}$ | $<100$ | m |
| Operating temperature range | $T_{op}$ | $-40$ to 80 | °C |
| Package size | | $\varphi 15 \times 32$ | mm |
| Weight | | $<6$ | g |

### 3.4. Validation

Before delving into the study of pipeline leakage acoustic signals, a preliminary test was performed to validate the reliability of the data collected. Two measurements were taken: first, the background noise of the experiment was recorded when there was no leakage, and the water pump was switched off. Second, with the pump activated but still no leakage, the noise produced by the high-pressure water pump was captured. Additionally, the acoustic signal from a leakage was recorded with an orifice size of 1 mm under an internal pressure of 0.2 MPa (refer to Figure 4). A comparative analysis of the data reveals certain patterns. The trends across the three recorded curves display a broad similarity: amplitudes are higher at lower frequencies and decrease as frequency rises. With the water pump off, the amplitude of the background noise is subdued, maintaining a consistent pattern. In contrast, the red curve representing the water pump noise shows an amplitude greater than the background noise, presenting distinct waveforms in the high-frequency spectrum. The leakage noise amplitude closely mirrors that of the water pump noise, especially in the high-frequency domain, where the two are nearly indistinguishable.

This suggests that the leakage noise is overshadowed by the noise from the water pump, rendering it an unreliable indicator of leakage signals.

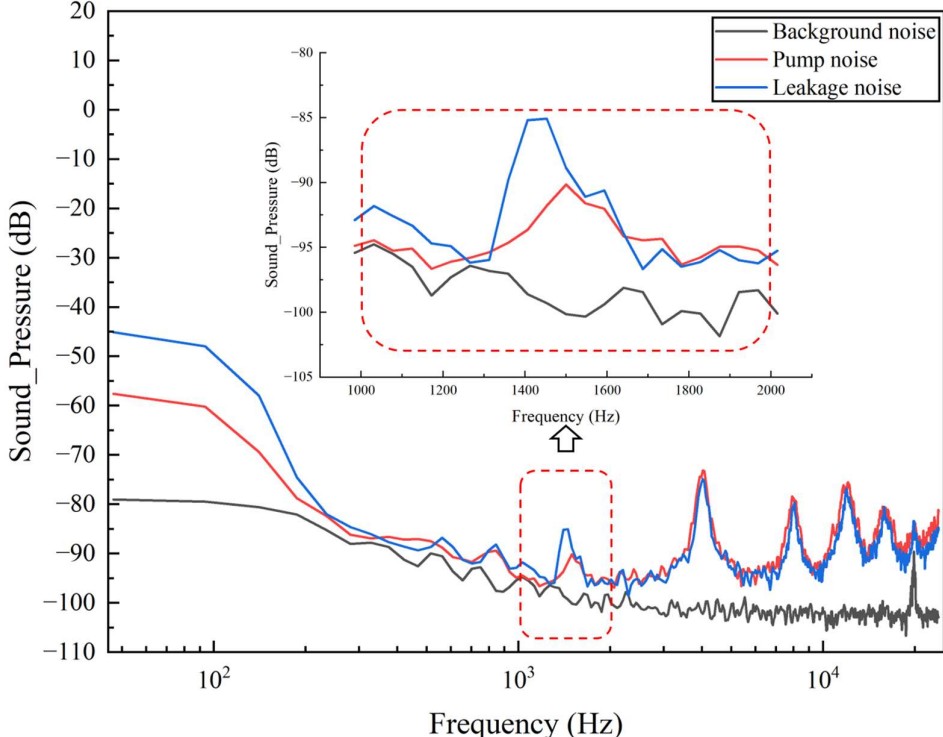

**Figure 4.** Experimental data for determining leakage characteristics (1 mm, 0.2 MPa).

In the frequency range between 1 kHz and 2 kHz, the leakage noise curve markedly diverges from the water pump noise curve. This suggests that this specific frequency range can serve as a distinguishing feature for leakage signals, a detail which is highlighted in Figure 4. This observation aligns with the findings of Xu et al. [28], who undertook detailed studies on the acoustic signals of pipeline leakages. Their experimental pipeline featured an inner diameter of 200 mm, and they operated within a pressure range below 1 MPa, making their experimental conditions similar to ours. Consequently, a frequency range from 1 kHz to 4.3 kHz was identified as the optimal band for extracting leakage signal characteristics. Figure 5 depicts the frequency-domain sound pressure level curve of pipeline leakage as determined from their experiment, given a 1 mm leakage aperture and a 0.2 MPa leakage pressure. For clarity, the conditions under which this leakage curve was produced involved a 1 mm leakage aperture and an internal pressure of 0.2 MPa, conditions that mirror those of the leakage curve in Figure 5. A side-by-side comparison reveals a consistent trend between the sound pressure level curves from both experiments as well as congruence in the frequency domains of the leakage sound characteristics. This validation underscores the precision and reliability of our experimental system and approach for in-depth analyses of pipeline leakage acoustic signals.

Xu et al. [28] conducted highly scientific research on the sound signals of pipeline leakage, examining how different leakage apertures and internal pressures affected the sound signal. However, there remains a scope for more extensive leakage experiments. This study explores a broader range of leakage aperture variations and more nuanced changes in internal pipe pressure. Additionally, we investigated the impact of detection distance on acoustic signal detection. As such, this research not only builds upon prior studies but also offers insights for forthcoming investigations.

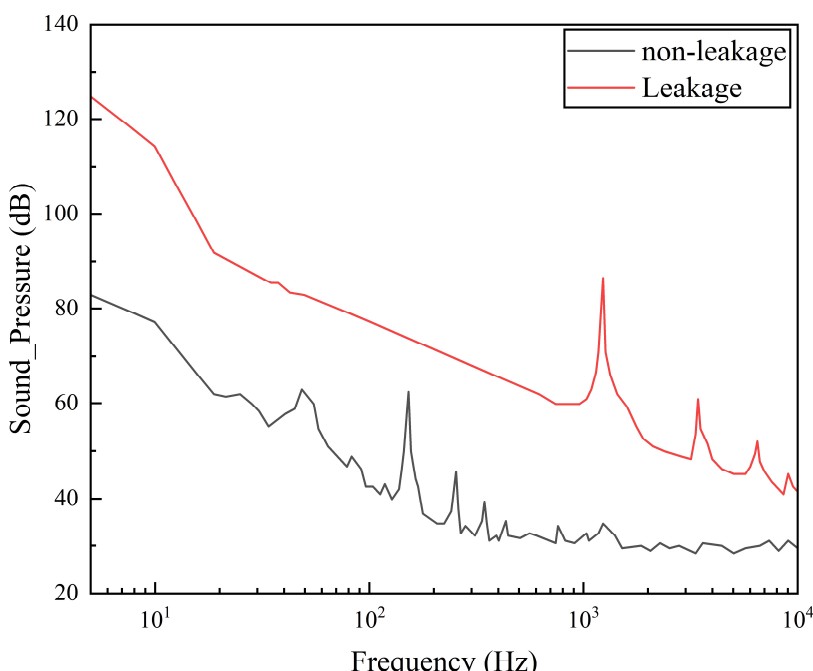

**Figure 5.** Experimental data from Tianjin University (1 mm, 0.2 MPa, Xu et al. [28]).

## 4. Experimental Data Processing

Research indicates that the sound waves produced by fluid leakage in pressurized vessels primarily stem from three sound sources: turbulent noise, bubble noise, and turbulent boundary layer pulsating pressure. Among these, turbulent noise is predominant, as supported by experimental data. This work delves into the near-field sound sources of leakage and the acoustic propagation characteristics during a pipeline's steady-state leakage phase. As a result, the detected leakage sound signals predominantly originate from the turbulence fluctuations of the leak jet. Within the leakage orifice, turbulent vortices fluctuate inconsistently, leading to momentary variations in the frequency-domain leakage sound signal curve. Thus, for experiments conducted under consistent conditions, it is imperative to sample the frequency-domain leakage sound signals multiple times and then average them. This process minimizes random discrepancies in the experimental data, ensuring accuracy. This article investigates leakage sound signals by manipulating three variables: the size of the leakage orifice, the pipe's internal pressure, and the axial distance between the leakage point and the hydrophone.

### 4.1. Influence of Pressure on Leakage Noise Signal

4.1.1. Acoustic Signal with Leakage Aperture of 0.5 mm

At axial distances of 0 mm, 500 mm, 1000 mm, 1500 mm, and 2000 mm from the hydrophone, the sealing plugs were sequentially replaced with leakage holes measuring 0.5 mm in diameter. To ensure only one active leakage point, the other five positions were sealed using threaded plugs. The experiment maintained a consistent pressure within the pipe using a high-pressure pump, a back-pressure valve, and a pulsation damper. The desired experimental pressure, ranging from 0.2 MPa to 1.0 MPa in 0.1 MPa increments, was set by adjusting the back-pressure valve. The measured data are presented in Figure 6, where the curves in the characteristic frequency range have been enlarged for detailed inspection. The experimental data of each group correspond to the respective detection distance positions in Figure 2.

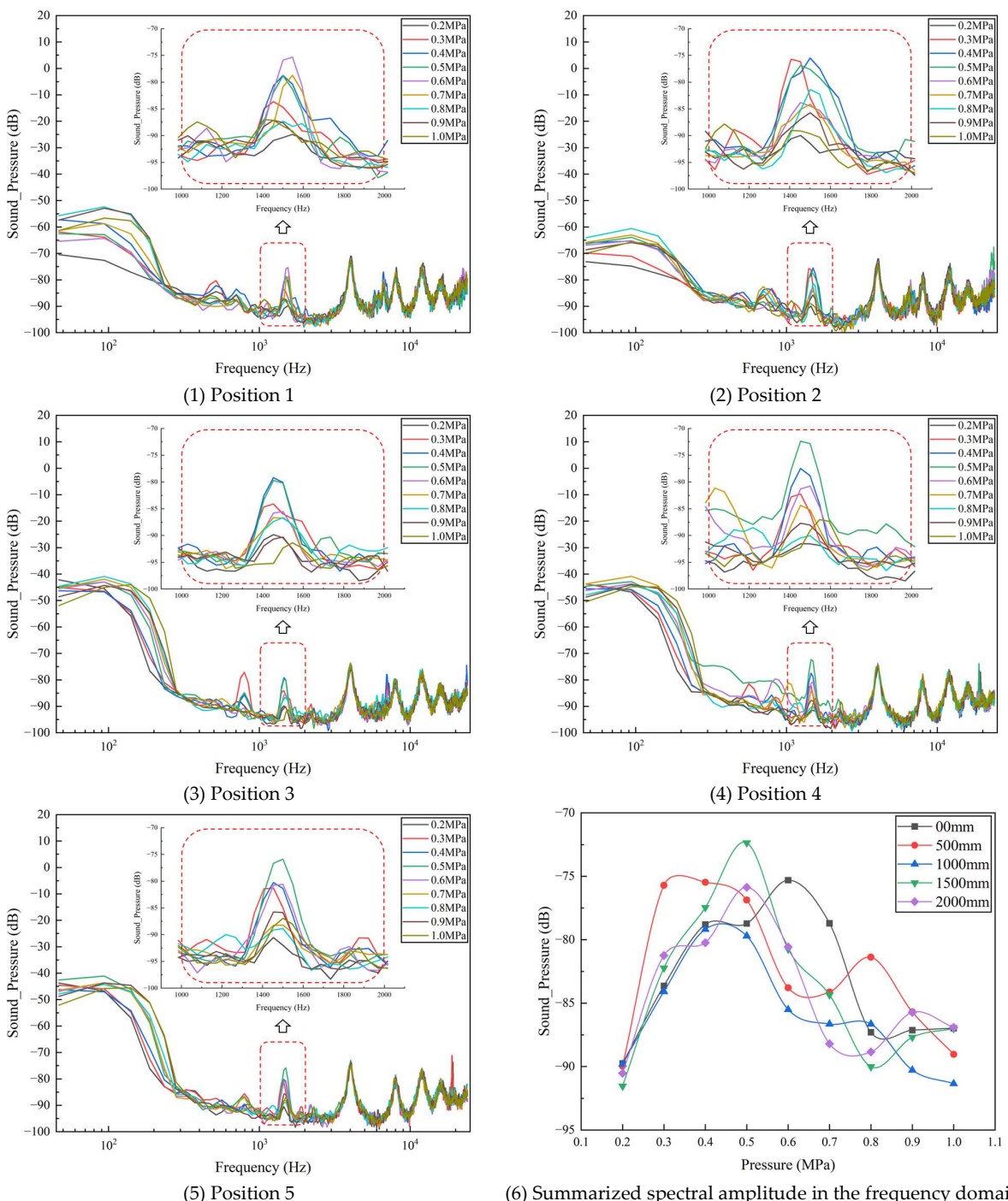

**Figure 6.** Acoustic signal curve and characteristic frequency band amplitude summary curve of leakage holes with a diameter of 0.5 mm measured by changing pressure at different positions.

Upon analyzing the peak values within the characteristic frequency range of our data, we obtained the characteristics of leak sound signals emanating from a 0.5 mm leak aperture at different positions and pressures. A clear observation from summarized data shows that as the leak pressure escalates, the sound signals from the 0.5 mm aperture, regardless of its position, consistently exhibit a trend: initially increasing and then decreasing. This pattern mirrors findings from previous studies. For instance, Wang et al.'s research on the sound signal of pipeline leakage and found that not all powers under different leakage conditions consistently increased with pressure [29]. This observed phenomenon is believed

to be influenced by the combined effect of the vortex magnitude in the leak hole and the boundary layer thickness within the leak hole.

To further understand the vortex conditions inside the leak hole during pipeline leakage, we modeled the leaking pipeline and used Ansys Fluent (2022 R1) to employ the large eddy simulation (LES) simulation calculation on the leaking flow field (Figure 7). The simulation revealed that for a fixed leakage aperture, greater internal pipe pressures result in more vortices forming at the leakage aperture. Therefore, there is a greater occurrence of distortions and breaks in these vortices. This predominates as the main reason the amplitude of the leakage sound signal within the characteristic frequency domain intensifies as the internal pipeline pressure rises.

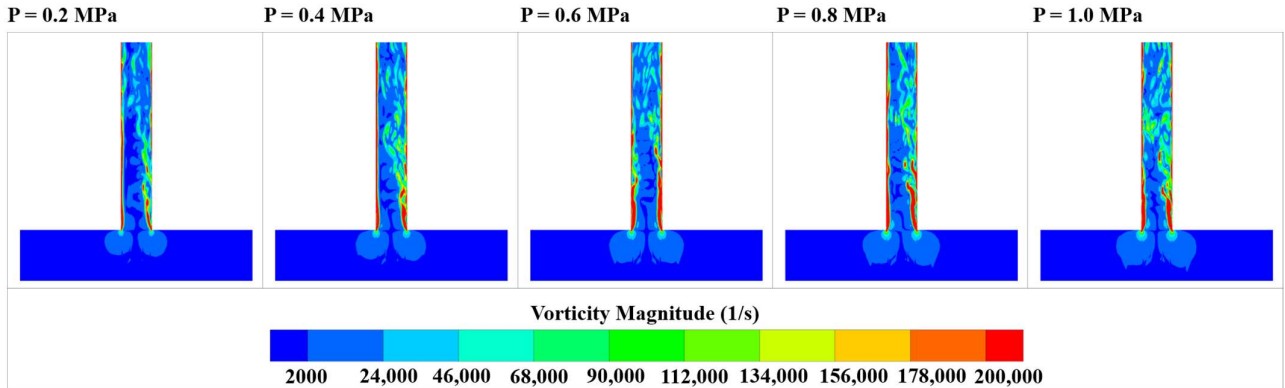

**Figure 7.** Simulation results of vorticity magnitude in leakage holes under different pressures.

In fundamental fluid mechanics, the boundary layer's thickness is associated with the Reynolds number, a dimensionless quantity in fluid mechanics to characterize the influence of viscosity. The equation for the Reynolds number is

$$\text{Re} = \frac{\rho v L}{\mu} \tag{7}$$

Here, $\rho$ denotes the fluid density, $\mu$ represents the coefficient of kinetic viscosity, $v$ denotes the characteristic velocity of the flow field, and L represents the characteristic length of the flow field. Physically, the Reynolds number represents the ratio of inertial force to viscous force. In cases of external flow, $v$ and L typically refer to the velocity of the far-ahead flow and the primary dimension of the object, respectively. For internal flows, they often represent the channel's average flow velocity and its diameter. A smaller Reynolds number highlights the prominence of viscous forces, while a larger one underscores the significance of inertia.

Under leak conditions, variations in the Reynolds number can alter the turbulent boundary layer's thickness. The equation is as follows:

$$\sigma = 0.37 \frac{1}{\text{Re}^{0.2}} \tag{8}$$

where $\sigma$ is the thickness of the turbulent boundary layer. Re denotes the Reynolds number. In these experiments, only the pipe's internal pressure was adjusted, with all other conditions being the same. As the internal pressure increases, so does the leakage amount, even though the leakage aperture remains unchanged. Therefore, $v$ increases in tandem with the increase in pressure inside the pipe. Given that fluid inside the pipe remains constant, $\rho$ and $\mu$ are constant. Similarly, the leakage hole has not been replaced, making L a constant as well. In summary, under the experiment's parameters, the thickness of the turbulent boundary layer decreases with increasing $v$ (see Figure 8).

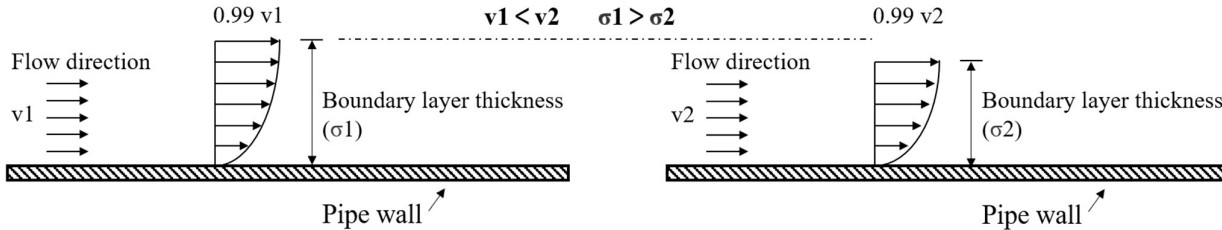

**Figure 8.** Relationship between fluid velocity and boundary layer thickness.

The intensity and spectral characteristics of turbulent sound are closely related to the thickness of the boundary layer. A thick boundary layer results in a more gradual change in fluid flow velocity, leading to the formation of larger vortices that persist for extended periods. The interactions between these large-scale vortices and the surrounding fluid primarily produce low-frequency turbulent sound. Thus, with a thicker boundary layer, there is a more pronounced presence of low-frequency turbulent sound. Conversely, a thin boundary layer is characterized by more dramatic changes in fluid flow velocity, resulting in the formation of smaller, closely packed vortices. These small-scale vortices generate and dissipate turbulent energy rapidly, contributing to the production of high-frequency turbulent noises. Hence, a thinner boundary layer leads to more significant high-frequency turbulent sounds. This phenomenon explains the decrease in leakage sound as the internal pipe pressure increases.

The combined influence of these factors means that as the pressure inside the pipe rises, the leakage sound first increases and then decreases. Between pressures of 0.2 MPa and 0.5 MPa, the influence of vortex size on leakage sound surpasses that of boundary layer thickness. In this range, the leakage sound intensifies as the internal pressure rises. However, between pressures of 0.5 MPa and 1.0 MPa, the effect of the boundary layer's thickness on the leakage sound becomes more dominant than that of vortex size, causing the leakage sound to diminish as the internal pressure continues to increase.

In our experimental setup, the sound signal amplitude within the characteristic frequency band reached its peak when the 0.5 mm leakage hole was subjected to a pressure of 0.5 MPa. This observation can offer valuable insights for pipeline detection.

4.1.2. Acoustic Signal with Leakage Aperture of 1.0 mm

For the subsequent experiment, we adjusted the leakage aperture to 1 mm. The results, illustrated in Figure 9, are noteworthy.

The trend derived from the 1 mm leakage aperture mirrors that of the 0.5 mm aperture—initially increasing and then decreasing. However, the overall amplitude of the signal from the 1 mm leak is noticeably higher. The most pronounced sound signal amplitude in the characteristic frequency band occurs when the internal pressure stands at 0.4 MPa.

From these two experiments, which solely varied the internal pipeline pressure, we deduce that for detecting minor leaks in pipelines, setting the internal pressure between 0.4 MPa and 0.5 MPa is optimal. This pressure range produces the most distinct pipeline leakage sound signals, enhancing detection accuracy.

*4.2. Influence of Aperture on Leakage Noise Signal*

Further experiments were carried out using leakage holes of varying diameters: 0.5 mm, 1.0 mm, 1.5 mm, 2.0 mm, and 2.5 mm. These were positioned at axial distances of 0 mm, 500 mm, 1000 mm, 1500 mm, and 2000 mm from the hydrophone. Only one position was active with a leakage hole, while the other positions were securely sealed with threaded plugs. To maintain a consistent internal pressure of 0.2 MPa, we utilized a high-pressure pump, a back-pressure valve, and a pulsation damper. The recorded data are presented in Figure 10.



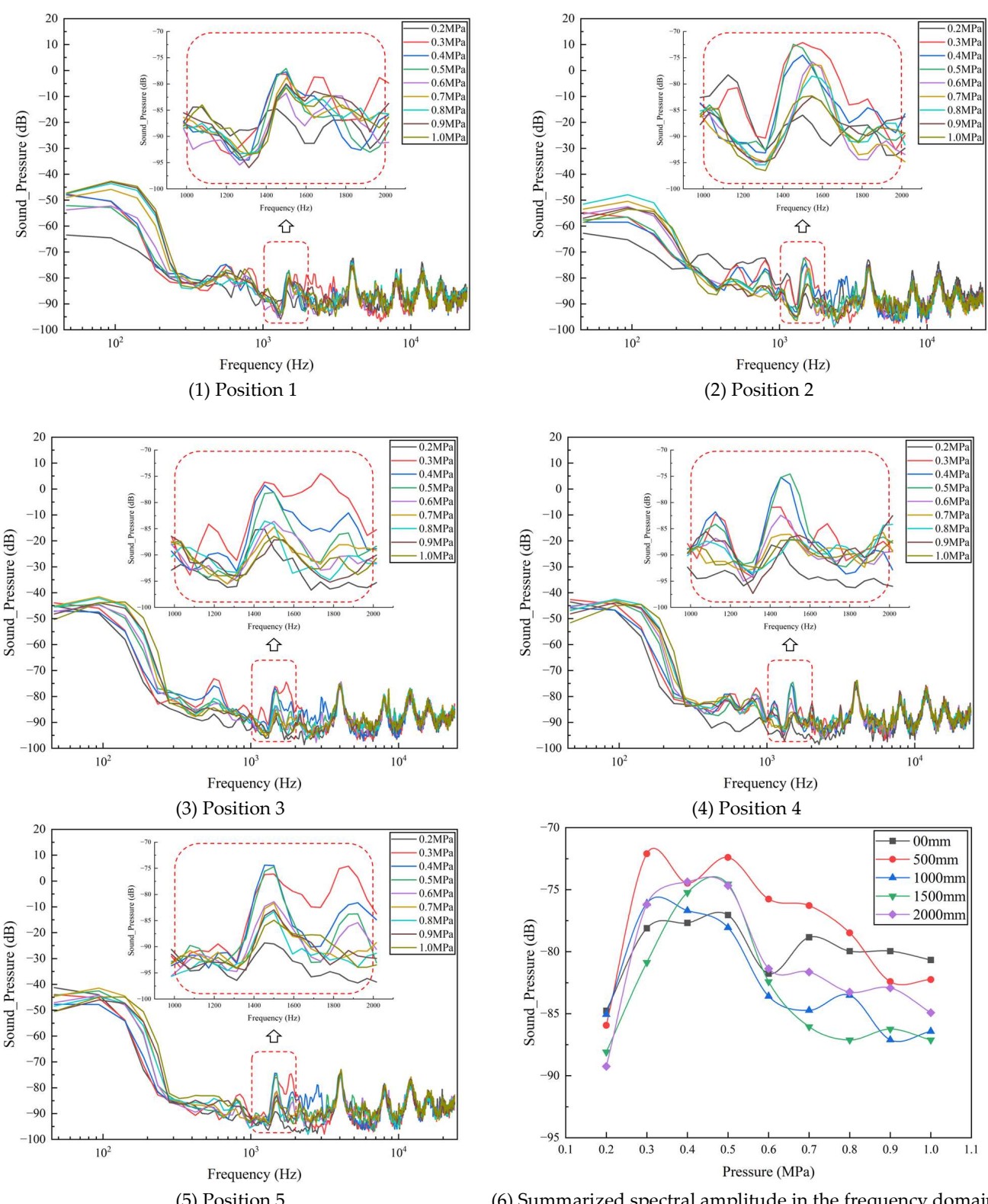

**Figure 9.** Acoustic signal curve and characteristic frequency band amplitude summary curve of leakage holes with diameter of 1.0 mm measured by changing pressure at different positions.

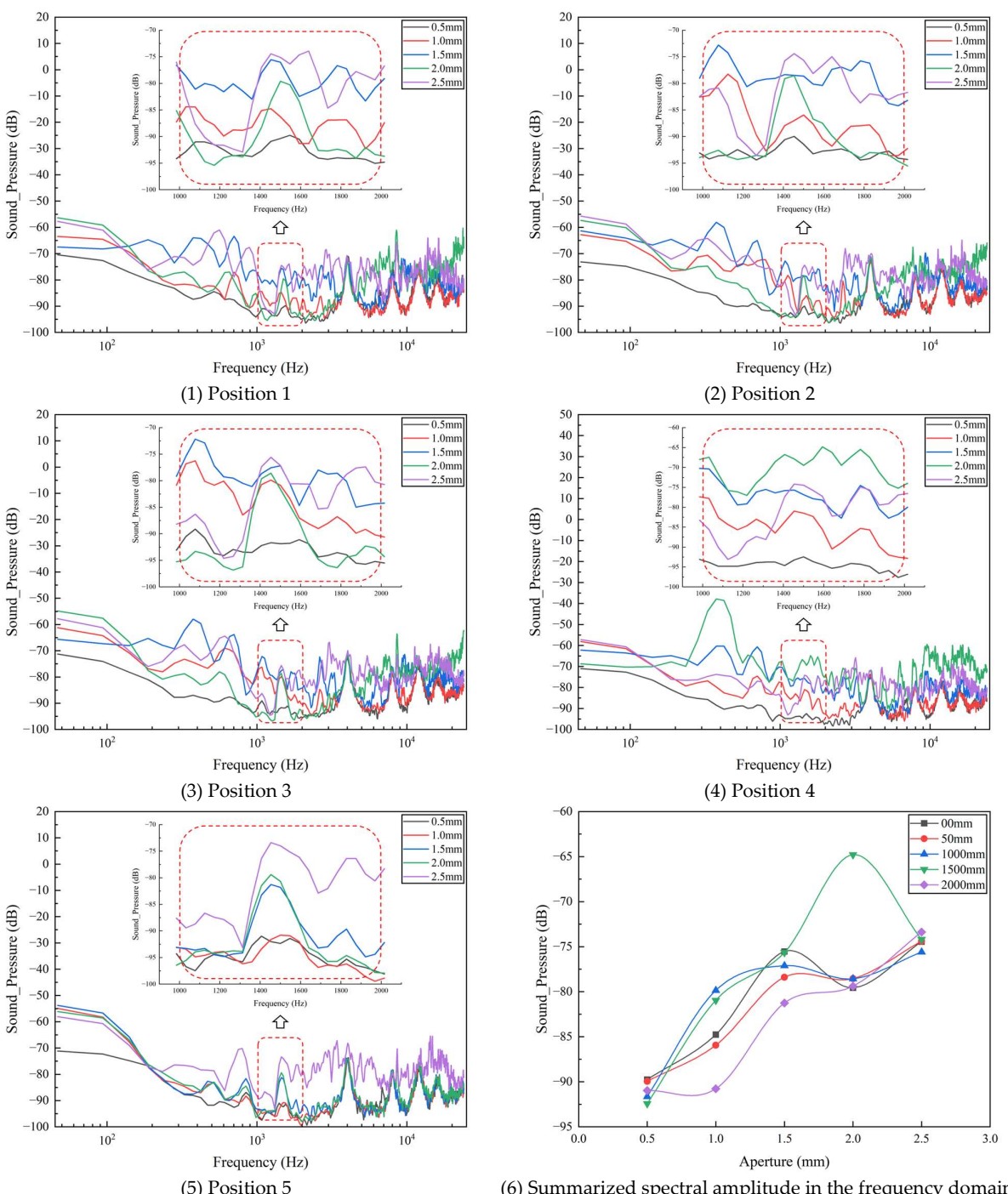

**Figure 10.** Acoustic signal curve with a leakage pressure of 0.2 MPa and the summary curve of the acoustic signal amplitude within the characteristic frequency were measured at different positions by changing the leakage aperture.

From our analysis of the summarized curves, it is evident that as the leak aperture enlarges, the amplitude of the leak sound signal correspondingly rises. Table 2 displays the leakage flow data acquired during the experiment, 1–3 means three groups of the experiment at the same condition. Upon analysis, a strong correlation emerges between the leakage flow rate and the cross-sectional area of the leak. This suggests that when the leakage pressure remains constant, the speed of the leak is relatively stable. As previously discussed, when this leakage velocity is unchanging, the boundary layer thickness

inside the leakage orifice remains consistent, exerting a negligible influence on the leak sound signal.

**Table 2.** Three experimental results on relationship between leakage flow rate and leakage aperture.

| Aperture (mm) | Hole Area (mm$^2$) | Flow (L/h) | Flow (L/h) | Flow (L/h) |
|---|---|---|---|---|
| | | 1 | 2 | 3 |
| 0.5 | 0.0625π | 15.2 | 15.0 | 14.5 |
| 1.0 | 0.2500π | 60.0 | 53.3 | 52.5 |
| 1.5 | 0.5625π | 116.5 | 112.5 | 113.0 |
| 2.0 | 1.0000π | 235.0 | 236.0 | 237.0 |
| 2.5 | 1.5625π | 360.0 | 357.0 | 360.0 |

Under these conditions, the vorticity within the leakage orifice becomes the primary determinant for fluctuations in the leak sound signal. We modeled pipeline models with different leakage apertures and used Ansys Fluent (2022 R1) for transient leakage flow field simulation calculations. Figure 11 depicts the magnitude of vorticity inside leak orifices of varying dimensions, all subjected to the same leakage pressure. One can infer that with the enlargement of the leakage aperture, both the scale and number of vortices have enhanced potential for development. This is attributable to the fact that larger leak orifices, by virtue of their greater diameter, allow the fluid to create more substantial and numerous vortex structures. Such conditions facilitate the detachment of vortices from the boundary layer and subject them to heightened pressure fluctuations. These fluctuations, stemming from distortion, deformation, and dissipation, amplify the turbulent sound. In essence, there is a positive relationship between the size of the leak and the intensity of the turbulent sound, with larger leaks yielding more pronounced turbulence sounds.

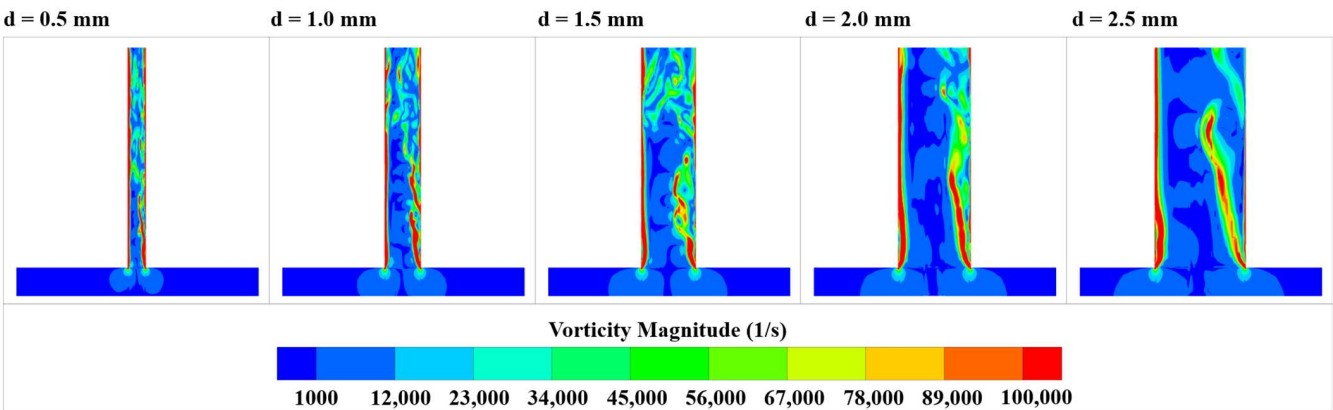

**Figure 11.** Simulation results of vorticity magnitude in leakage holes with different apertures.

The findings from this experiment provide valuable insights for practical engineering procedures. When performing leak inspections on pipelines, the intensity of the leak sound signal can offer clues about the size of the leak.

### 4.3. Influence of Detection Distance on Leakage Noise Signal

The experimental setup mirrors that detailed in Section 4.2, so it will not be repeated here. Instead, this section delves into how detection distance influences the leak sound signal. We measured the leak sound signal at axial distances of 0 mm, 500 mm, 1000 mm, 1500 mm, 2000 mm, and 2500 mm from the hydrophone. During the experiment, the pressure inside the pipeline was 0.2 MPa. Detailed data can be found in Figure 12.

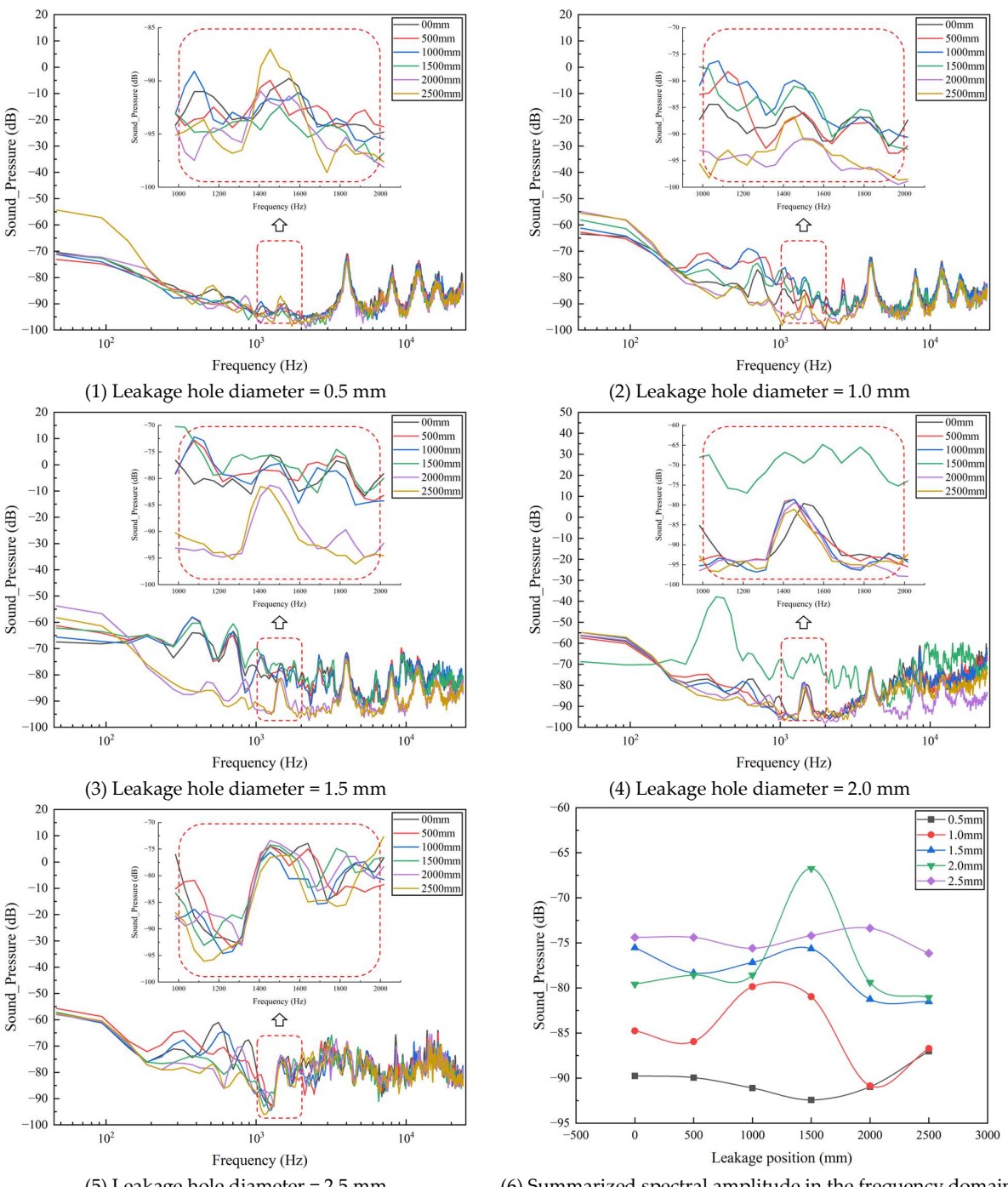

**Figure 12.** By changing leakage position and measuring acoustic signal curve with leakage pressure of 0.2 MPa under different aperture conditions, summary curve of acoustic signal amplitude within characteristic frequency is obtained.

A review of the experimental data reveals that, throughout the experiment, the amplitude of the leak sound signals taken at six varying positions remained largely consistent. In this set of experiments, while the detection distance was modified by altering the leak's position, the results did not display any pattern of the leak sound signal increasing as the detection distance narrowed. Several factors might account for this observation: Firstly, given the limited length of the experimental pipeline, the sound signal originating from the leak point was in close proximity to the hydrophone, leading to negligible attenuation of

the leak sound signal during its transmission. Secondly, the characteristics of the experimental pipeline material, along with its internal surface characteristics, might influence the transmission of sound signals, ensuring that the amplitude of the leak sound signal stays relatively constant.

In real-world pipeline operations, when the pipeline is long, the acoustic signal from a leak will diminish and eventually fade as it travels along the pipeline (see Figure 13). During pipeline inspections, the detector initially picks up a faint leakage sound signal. However, as the detector approaches the source of the leak, the received sound signal becomes more pronounced. This progressive intensification aids in determining the exact location of the leak.

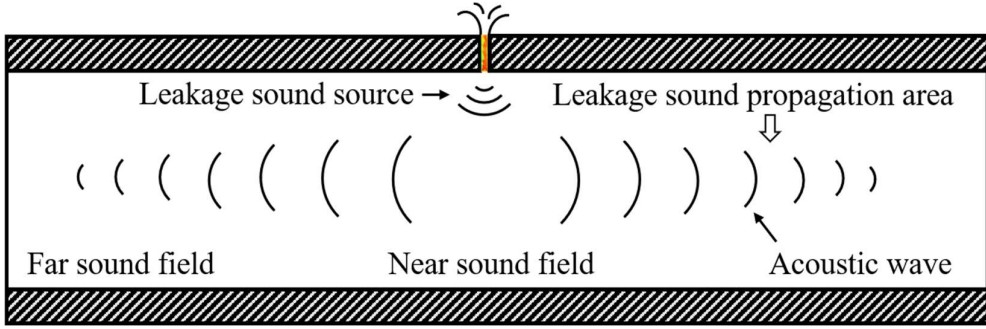

**Figure 13.** Schematic diagram of relationship between leakage sound intensity and propagation distance in reality.

## 5. Conclusions

This paper studies the characteristics of pipeline leakage sound sources and their near-field propagation characteristics through experimental methods. We designed a pipeline leakage experimental system and, in comparison with other studies, verified that the characteristic frequency domain of the leakage sound signal ranges from 1 kHz to 2 kHz. On this basis, we carried out multiple experiments, varying parameters such as internal pipe pressure, leakage aperture size, and detection distance, drawing the following conclusions:

(a)    Keeping all other variables constant, as the leakage pressure increases, the amplitude of the leakage sound signal in the characteristic frequency band first increases and then decreases. In the context of our experiment, the leakage sound signal is most pronounced at pressures between 0.4 and 0.5 MPa. This finding is significant for practical engineering operations. For specific pipeline inspections, it is advisable to adjust the internal pressure to fall within the 0.4 MPa to 0.5 MPa range. At this pressure level, detectors within the pipeline can more easily identify sound signals from smaller leaks, thus preventing more severe leakage accidents.

(b)    With other conditions held constant, increasing the size of the leakage aperture leads to a steady rise in the leakage sound signal within the characteristic frequency band. This insight is also of practical relevance. Under uniform inspection conditions, the intensity of the leakage sound signal collected by the internal pipeline detector can be used to determine the degree of pipeline leakage.

(c)    Holding other variables constant, changing the distance between the hydrophone and the leakage hole does not lead to significant changes in the acoustic signal within the characteristic frequency band. This phenomenon can be attributed to the brevity of our experimental pipeline and the minimal attenuation of sound signals across short distances, which influenced our results.

In summary, this article supplements previous research and offers valuable insights for practical engineering operations. By applying the conclusions of this study, it can contribute to the timely detection of small leaks in pipelines and the prevention of leakage accidents.

However, due to limitations in the experimental system conditions, there is still a shortcoming in this study. The leakage testing pipeline is relatively short, which is insuffi-

cient to reflect the impact of detection distance on acoustic signals, and thus the amplitude of the leak sound signal stays relatively constant with the change in the detecting distance.

**Author Contributions:** Conceptualization, K.Z.; methodology, K.Z.; software, K.Z. and R.M.; validation, K.Z., R.M. and T.G.; investigation, K.Z.; writing—original draft preparation, K.Z.; writing—review and editing, T.G., J.Y. and Y.G.; supervision, K.Z. and Y.G.; funding acquisition, K.Z. All authors have read and agreed to the published version of the manuscript.

**Funding:** This research was funded by the National Natural Science Foundation of China, grant number U1908228, and the Fundamental Research Funds of the National Centre for International Research of Subsea Engineering and Equipment, grant number 3132023361.

**Institutional Review Board Statement:** Not applicable.

**Informed Consent Statement:** Not applicable.

**Data Availability Statement:** The data presented in this study are available on request from the corresponding author.

**Conflicts of Interest:** The authors declare no conflict of interest.

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
