# Peer review of "Experimental Investigation on Near-Field Acoustic Propagation Characteristics of Leakage Detection in Submarine Pipelines"

_jmse, doi:10.3390/jmse11102012_

Round 1
Reviewer 1 Report
Kindly read the attached file, thank you.

Author Response
Dear Editors and Reviewers:
The authors thank the Editor and all the reviewers for their time to review the manuscript. Following your suggestions, we have revised the manuscript (ID: jmse-2643112) entitled "Experimental investigation on near-field acoustic propagation characteristics of leakage detection in submarine pipelines".
The paper has been revised to improve the quality of the paper based on the comments received. A track changes version with changes shown in red font has been uploaded to highlight the changes.
We hope the quality of this revised paper could meet the publication requirement of this journal, and deeply appreciate your consideration of our manuscript. If you have any queries, please don’t hesitate to contact me at the address below.
Yours sincerely,
Dr. Kang Zhang
Dalian Maritime University
No.1 Linghai Road, Dalian, Liaoning province, China, 116026
E-mail: zhangkang@dlmu.edu.cn

Reviewer 2 Report
This study investigates the characteristics of pipeline leakage noise sources and the propagation properties of such noise in the near field. The formation mechanism of leakage sound sources was introduced, and corresponding theoretical research reviewed. The effects of pipeline pressure, leakage aperture, and detection distance on the acoustic signal characteristics are examined. The results show that as internal pipe pressure increases, the leakage sound signal intensity first increases and then decreases. As the leakage aperture increases, the intensity of the leakage sound signal increases. Within a short distance, the intensity remains consistent regardless of detection distance.
The shortcomings and missings of the paper are the following:
1. Some abbreviations are not decrypted, for example: PIG (L58), N-S (L142), FW-H (L156).
2. xi and xj (L151), Vi and Vj (L154) must be separated by commas.
3. The formulae of other authors must be accompanied by corresponding references (for example, formulae (4) and (5)).
4. Formula (4): Where is the decryption of parameters ui, f, p?
5. Formulae (5) and (6): What is FFT?
6. Table 1, penultimate line: j must be replaced by Æ.
7. Where is the reference on Fig. 7 into text? How were the modeling results obtained?
8. How were the modeling results obtained in Fig. 11?
9. How were Equations (1) - (6) used in the paper and if they were not used, why are they given?
10. Reference List breaks rule for papers and all papers must be accompanied by doi.
Author Response

(The authors gave the same response as above.)

Round 2
Reviewer 2 Report
The authors introduced necessary corrections and the paper could be accepted in this form.